# Correlation of the Abbe Number, the Refractive Index, and Glass Transition Temperature to the Degree of Polymerization of Norbornane in Polycarbonate Polymers

**DOI:** 10.3390/polym12112484

**Published:** 2020-10-26

**Authors:** Noriyuki Kato, Shinya Ikeda, Manabu Hirakawa, Hiroshi Ito

**Affiliations:** 1Mitsubishi Gas Chemical Company, 2-5-2 Marunouchi, Chiyoda-ku, Tokyo 100-8324, Japan; shinnya-ikeda@mgc.co.jp (S.I.); manabu-hirakawa@mgc.co.jp (M.H.); 2Graduate School of Science and Engineering, Yamagata University, 4-3-16 Jonan, Yonezawa, Yamagata 992-8510, Japan; ihiroshi@yz.yamagata-u.ac.jp; 3Graduate School of Organic Materials Science, Yamagata University, 4-3-16 Jonan, Yonezawa, Yamagata 992-8510, Japan

**Keywords:** polycarbonates, optical properties, thermal properties

## Abstract

The influences of the average degree of polymerization (*D_p_)*, which is derived from *M_n_* and terminal end group, on optical and thermal properties of various refractive indexed transparent polymers were investigated. In this study, we selected the alicyclic compound, Dinorbornane dimethanol (DNDM) homo polymer, because it has been used as a representative monomer in low refractive index polymers for its unique properties. DNDM monomer has a stable amorphous phase and reacts like a polymer. Its unique reaction allows continuous investigation from monomer to polymer. For hydroxy end group and phenolic end group polymers, the refractive index (*n_d_*) decreased with increasing *D_p_*, and both converged to same value in the high *D_p_* region. However, the Abbe number (*ν_d_*) of a hydroxy end group polymer is not dependent on *D_p_*, and the *ν_d_* of a phenolic end group polymer is greatly dependent on *D_p_*. As for glass transition temperatures (*T_g_*), both end group series were increased as *D_p_* increased, and both converged to the same value.

## 1. Introduction

The digitization of optical devices in recent years has greatly expanded the market for digital cameras. With the advent of camera-equipped mobile phones and smartphones, digital cameras have become ubiquitous. Due to their exceedingly small size and light weight, and its compatibility with the injection molding process for mass production, the material used in the camera is plastic rather than glass [1,2,3].

These cameras consists of a combination of lenses made of high refractive index and low Abbe number material and a lens made of low refractive index and high Abbe number material. The Abbe number, *v_d_*, has been used as a parameter for the wavelength dispersion of the refractive index, which is defined by the following equation:
*v_d_* = (*n_d_* − 1)/(*n_F_* − *n_C_*)
where *n_d_*, *n_F_*, and *n_C_* are the refractive indices of the material at the wavelengths 587.6, 486.1, and 656.3 nm.

The combination of these two materials is essential for correcting optical aberrations and obtaining high resolution images. To achieve high resolution cameras, the development of these high-refractive index and low-refractive index materials has been carried out. In addition, the demand for precision in the refractive index and Abbe number of materials is becoming more stringent with the increase in resolution.

The molecular design of optical polymer is based on the Lorentz-Lorenz equation, which defines the relationship between refractive index and polymer structure. According to this equation, refractive indices of polymers were determined by atomic refraction and molecular volume. Commercially available polymers with high refractive index include polycarbonates, polyesters, and polyester carbonate using bisphenol-A and 9,9-bis[4-(2-hydroxyethoxy)phenyl]fluorine (BPEF) as monomers. These monomers consist of aromatic rings having high atomic refraction, which realize a high refractive index (*n_d_* 1.58–1.64) [2,4,5,6,7,8]. On the other hand, commercially available polymers with low refractive index lens includes cyclo-olefin polymers (COP) and cyclo-olefin copolymers (COC) [9,10,11]. These polymers are mainly composed of the norbornane skeleton, an alicyclic hydrocarbon with relatively low atomic refraction, and have a refractive index of about 1.52–1.54 [12,13,14].

These days, polycarbonate resins using the alicyclic structure, dinorbornane dimethanol (DNDM) as a monomer have been developed [15]. The structure of DNDM is identical to the characteristic repetitive units of COP and COC. The polycarbonate using DNDM as a monomer shows relatively low *n_d_* and relatively high *ν_d_*. Therefore, lenses made of this resin are combined with lenses made of a material with a high refractive index to make up a camera lens. As mentioned above, demands for precision in refractive indices are becoming increasingly stringent. Since it is reported that the refractive index of high refractive index materials varies with the number of polymerization, it is also necessary to investigate low refractive index materials.

Previously, we investigated the relationship between *n_d_* and *D_p_*, and between *T_g_* and *D_p_*, about the high *n_d_* materials such as BPEF [16] and Bisphenol-A [17] based Polycarbonate polymers [18,19,20], which are materials for optical lenses. In the case of BPEF, *n_d_* decreased with increasing *D_p_* for hydroxyl terminated polymer and, whereas *n_d_* increase with increasing *D_p_* for phenolic terminated polymer. On the other hand, as for Bisphenol-A, *n_d_* decreased with increasing *D_p_* regardless of the end structures. Since the tendency of the refractive index changes with monomers, polycarbonate with DNDM as a monomer, a low refractive index material, should also be investigated.

For this reason, this study investigates the influence of the degree of polymerization and end structures of polymer to the optical and thermal properties of the alicyclic DNDM polycarbonate.

## 2. Experimental Section

### 2.1. General

Dinorbornane dimethanol (DNDM), manufactured by Mitsubishi Gas Chemical Company, Inc. (Tokyo, Japan), was used without further purification. Other chemicals at reagent grade were also used without further purification. Refractive indices and Abbe number of oligomers and polymers were measured, using a Kalnew precision refractometer KPR-3000 (manufactured by Shimadzu, Kyoto, Japan). Molecular weights and molecular weight distributions were estimated by gel permeation chromatography using the Shodex GPC-101 high speed liquid chromatography system (SHOKO SCIENCE Co., Ltd., Yokohama, Japan) equipped with three consecutive Shodex LF804 columns (Showa Denko K.K., Tokyo, Japan) eluted with THF that was calibrated by polystyrene standards (EsaiCal PS-2). Differential scanning calorimetry (DSC) was performed by X-DSC7000 (Hitachi High-Tech Science Corporation, Tokyo, Japan) at a heating rate of 10 K/min and at a cooling rate of −2 K/min under N_2_ atmosphere at a flow rate of 50 mL/min. The glass transition temperature was taken as the temperature in the middle of the thermal transition from the second heating scan.

### 2.2. Monomer Synthesis

DNDM, the monomer investigated in this study, was synthesized by performing a Diels-Alder reaction, an Oxo reaction, and a Hydrogenation reaction in sequence. At first, dicyclopentadiene was heated and turned into cyclopentadiene. Subsequently, cyclopentadiene and methyl acrylate were synthesized by the Diels-Alder reaction to form norbornene methyl esters. A dinorbornene methyl ester was obtained by further adding cyclopentadiene to a norbornene methyl ester, and an oxo reaction was carried out using a synthesis gas, followed by hydrogenation under conditions of high temperature 488 K and high pressure 10 MPa to obtain DNDM. Finally, distillation was performed to obtain purified DNDM. Taking advantage of the absence of aldehyde regioselectivity for the oxo reaction, the ratio of 2,6- and 2,7-isomers were 50% of DNDM, respectively. DNDM is also synthesized in mixtures to eliminate crystallinity. Thereby, DNDM became a mixture without crystallinity and a DNDM polymer having extremely high amorphous property was obtained. This high amorphous property is crucial during the injection molding process. Because fisheye and white dot phenomena such as cyclo-olefin polymers are less likely to occur, it is possible to perform molding under a wider range of molding conditions in this occasion.

For the sake of simplification, DNDM structures will hereinafter be described as shown in Figure 1:

Figure 2 shows the DSC charts of the DNDM monomer. During scanning, a melting point was not found, but a glass transition temperature (*T_g_*) was observed at 291.8 K. This indicates that the DNDM monomer possesses a stable amorphous phase and acts as if it were a polymer. The low molecular weight DNDM polycarbonate oligomers, such as dimer and trimer, also follow this feature and possess the *T_g_*. This interesting feature allows for continuous investigation of optical and thermal properties of DNDM polycarbonate (DNDM-PC) from monomer to oligomer, and polymer.

### 2.3. Oligomer and Polymer Synthesis

All polycarbonate oligomers and polymers containing DNDM were synthesized by the transesterification method in “good to excellent” yields (Scheme 1). That is, the thermal reaction of DNDM with diphenyl carbonate (DPC) in the presence of a catalytic amount of NaHCO_3_ afforded DNDM-PC. Simultaneously, phenol as a by-product was removed under reduced pressure to shift the equilibrium towards the polymerization side.

Degree of polymerization (*D_p_*) and terminal end group was easily and precisely controlled by a selection of the ratio of DNDM and DPC. For example, DNDM dimer bearing hydroxy end group (OH-2) was synthesized by the transesterification reaction with the ratio of 2:1 of DNDM/DPC. On the other hand, DNDM dimer bearing phenolic end groups (Ph-2) was obtained by the reaction with the ratio of 2:3 (Scheme 2):

## 3. Results and Discussions

All fundamental properties, thermophysical properties, as well as optical properties are summarized in Table 1 and Table 2 [21].

*D_p_* is calculated by the formula below:

Hydroxy end group;
*D_p_* = (*M_n_* + *a* − 2*c*)/(*b* + *a* − 2*c*)(1)

Phenolic end group;
*D_p_* = (*M_n_* − *a* − 2*d*)/(*b* + *a* − 2*c*)(2)
where:*a* is the molecular weight of carbonyl groups; 28.01.*b* is the molecular weight of DNDM; 222.33.*c* is the molecular weight of hydrogen; 1.01.*d* is the molecular weight of benzene ring; 93.11.

### 3.1. Relation between n_d_ and D_p_

As necessary, the specimens for refractive index were prepared by two different methods. As for the oligomers having enough viscosity, 40 mm diameter and 3 mm thick circular plates were prepared by a press molding and then cut to form right angles. Then, the cut surface of specimens were polished using REFINE POLISHER RPO-128B and were subjected to mirror finishing. Regarding the other specimens with low viscosity, the molten oligomers were poured into prism- shaped molds and cooled until solidified. The resulting specimens were measured without further polishing. The refractive indices of resulting specimens were measured by Kalnew KPR-3000. Figure 3 shows the relationship between refractive index (*n_d_*) at d-line (587.6 nm) and *D_p_* for DNDM homo polymer series with the hydroxyl end group and the phenolic end group.

With respect to the hydroxyl end series, the dimer (OH-2) showed high *n*_d_ of about 1.5467. As *D_p_* increases, the *n_d_* value decreases as 1.5431 (OH-3), 1.5337 (OH-10), 1.5323 (OH-17), and 1.5299 (OH-31). For phenolic end series, *n_d_* value of dimer (Ph-1) was 1.5529. As *D_p_* increases, *n_d_* value also decreases as 1.5490 (Ph-2), 1.5390 (Ph-8), 1.5339 (Ph-18), 1.5326 (Ph-28-2). Almost the same tendency was observed regardless of the difference in the end group. The *n_d_* value of phenolic end series is higher than that of the hydroxy end series in the region of *D_p_* from 1 to 20. However, the difference between the hydroxyl end series and the phenolic end series is negligibly small and *n_d_* converges to almost the same in a sufficiently large *D_p_* region. Although the refractive index of polymers is generally regarded as intrinsic, *n_d_* of polymers depends on *D_p_* as previously revealed. In addition, herein, *n_d_* of polycarbonate using alicyclic monomer, DNDM, also showed dependency to *D_p_*. Thus, the refractive index of polycarbonate seems to depend on the *D_p_* and the end group of the polymer, whether the constituent monomers are aromatic or aliphatic.

Furthermore, to discuss the hydroxyl end group and phenolic end group from the same perspective, we focused on the weight percentage of carbonyl groups (*CO*). *CO* against whole molecule for hydroxyl end group was calculated by the following formula:
*CO* = *a* (*D_p_* − 1)/(*b D_p_* + *a* (*D_p_* − 1) − 2 *c* (*D_p_* − 1)) × 100(3)
where:*CO* is weight percentage of carbonyl groups against whole molecular weight.*D_p_* is degree of polymerization.*a* is molecular weight of carbonyl groups, 28.01.*b* is molecular weight of DNDM; 222.33.*c* is molecular weight of hydrogen; 1.01.

In the case of the phenolic end group, *CO* was estimated by the following formula:
*CO* = *a* (*D_p_* + 1)/(*b D_p_* + *a* (*D_p_* + 1) − 2 *c D_p_* + 2 *d*) × 100(4)
where:*CO* is weight percentage of carbonyl groups against whole molecular weight.*D_p_* is degree of polymerization.*a* is molecular weight of carbonyl groups, 28.01.*b* is molecular weight of DNDM; 222.33.*c* is molecular weight of hydrogen; 1.01.*d* is molecular weight of benzene ring; 93.11.

From these equations, the relation between *D_p_* and the weight percentage of carbonyl groups *CO* is illustrated in Figure 4.

As *D_p_* increased, *CO* in the hydroxyl end series and the phenolic end series gradually approached 11.3%. With the above formulas, the relationship between *n_d_* and *CO* was investigated.

As a result, *n_d_* and *CO* were found to be represented by the relationship of quadratic polynomials within the scope investigated in this study, as shown in Figure 5, and is expressed in the following formula:*n_d_* = *α* × *CO^2^* + *β* × *CO* + *γ*(5)
where:*CO* is the weight percentage of carbonyl group against whole molecular weight.*α* is the constant intrinsic to the molecule.*β* is the constant intrinsic to the molecule.*γ* is the constant intrinsic to the molecule.

The α, β, and γ values for hydroxyl end series are −0.0004, 0.0036, and 1.5386, respectively, and those for phenolic end series are −0.0260, 0.6382 and −2.3619, respectively. From these coefficients, the intersection point of these two second order polynomial can be calculated. The intersection should show the intrinsic values of *CO* and *n_d_* for the DNDM polycarbonate polymer having infinite *D_p_*, and that is 11.26 and 1.5288, respectively.

This is the significance of converting *D_P_* to *CO*. Polycarbonate resin is composed of monomer and carbonate moiety as a linker. In the hydroxy end series, *CO* increases as the number of polymerization increases, and *CO* gets close to a specific limit. In the phenolic end series, *CO* is decreased by increasing the number of polymerization and *CO* gets close to a specific limit. Thus, *D_p_* goes to infinity, but *CO* does not, and converges to a certain value theoretically. Therefore, the properties of the critical molecular weight can be easily predicted by synthesizing the resins at several points in each hydroxy- and phenol-end series and measuring (the intersection point of Quadratic curve s of the hydroxy end series and the phenolic end series). Based on the facts specified above, it is useful to use *CO* instead of *D_p_* when defining the properties of polymers from an industrial perspective.

Additionally, because of the molecular weight dependence of the refractive index, the setting of the molecular weight range is important from the point of view of stability of the refractive index to the requirements of the optical products.

### 3.2. Relation between v_d_ and D_p_

Since it was found that the refractive index is affected by the number of polymerization and terminal end group, the dependence of the Abbe number (*ν_d_*) [22], which is a wavelength dispersion parameter of the refractive index, on the number of polymerization was also investigated.

With respect to the hydroxyl end series, the Abbe numbers were almost constant regardless of the increase in the number of polymers. For example, *ν_d_* values of OH-2, OH-14, and OH-31 are 57.5, 57.9, and 57.6, respectively. In contrast, for the phenolic end series, the *ν_d_* increased with the increase in *D_p_*, as in 46.1 (Ph-1), 55.2 (Ph-13), and 57.9 (Ph-28-1). In the region where the number of polymerization is sufficiently high, the Abbe number of both the phenolic and hydroxy end series converges to almost the same value. To the best of our knowledge, there is no example to investigate the influence of the number of polymerizations and end groups on the Abbe number in polymers.

Furthermore, the relationships between *CO*, which is derived from *D_p_*, and Abbe number *v_d_*, were examined, as shown in Figure 6. As analogous to the comparison between *ν_d_* and, *D_p_*, the behavior of *v_d_* versus *CO* is also different depending on the type of end group. In the case of the hydroxyl end group, *ν_d_* is almost constant regardless of *CO*, whereas in the phenolic end group, the *ν_d_* decreases linearly with increasing *CO*. Since the Abbe number is a measure defined by *n_d_*, *n_F,_* and *n_C_*, the correlation between *CO* and *n_d_*, *n_C_*, and *n_F_* was investigated. The results were shown in Figure 7 and Table 3.

As with the relationship between *n_d_* and *CO*, the relationship between *n_F_* and *CO*, as well as the relationship between *n_C_* and *CO* could be also approximated by quadratic polynomials.

In the case of the hydroxyl end group, the shape of the quadratic curves in *n_d_*, *n_C_*, and *n_F_* are almost identical, and the coefficients of the quadratic term in the quadratic polynomial are also almost the same. As a result, the values of *ν_d_*, as defined by *n_d_*, *n_C_* and *n_F_*, are constant for the whole measured region of *CO*. On the other hand, the situation is different for the phenolic end group. The curvatures of quadratic curves of *n_d_*, *n_C_*, and *n_F_* are different from each other, and the coefficients of quadratic term of the quadratic polynomials are −0.0260, −0.0238, and −0.0279, respectively. This indicates that the change in refractive index in relation to the change in *CO* is large in the order of *n_F_*, *n_d_*, and *n_C_*.

As a results, relatively large changes in *n_F_* in response to changes in *CO* lead to a large dependence of the Abbe number on *CO*. The details are unclear, but the molecular end groups may be affected to the *ν_d_*. The Abbe number, *ν_d_*, is also defined by Equation (6):(6)$$\nu_{d}\;=\;\frac{6n_{d}}{\left(n_{d}{}^{2}+2\right)\left(n_{d}+1\right)}\frac{\left[R\right]}{\left[\Delta R\right]}$$
where [*R*] and [∆*R*] are molar refraction and molar dispersion, respectively. According to Equation (6), when ∆*R* becomes larger, *ν_d_* becomes smaller. The phenyl group is known to have a large molar dispersion [23], and the effect of the terminal phenolic group is more pronounced in the region of small polymerization numbers for a series of phenolic end group. As the number of polymerization increases, the influence of the phenolic end group decreases. Eventually, the effect of the phenolic end group becomes negligible and approaches a certain value. On the other hand, in the case of the hydroxyl end group, there are no phenyl groups with high atomic dispersion. Since the molecules are composed only of the DNDM and carbonate bonds, the value of *ν_d_* is nearly constant in the measured region. Because of the large dependence of *ν_d_* on molecular weight and end groups, the control of the molecular weight and selection of the end groups are very important from the point of view of the optical stability of the product.

The relationships between *CO* which is derived from *D_p_* and Abbe number *v_d_* were examined, as shown in Figure 6. There are no other examples, to the best of our knowledge, that reveal the relationship between *D_p_* and *v_d_* and end groups in aliphatic polymers.

Notably, the behavior of *CO* versus *v_d_* is quite different depending on the type of terminal end group. In the phenolic end group, the *v_d_* decreases with increasing *CO*, whereas the hydroxy end group, *v_d_* is almost constant regardless of *CO*.

For analysis, we described the correlation of *CO* and *n_d_*, refractive index (*n*_C_) at C-line (656.3 nm), and refractive index (*n*_F_) at F-line (486.1 nm)*,* shown in Figure 8. The correlation of the refractive indices and *CO* is shown in Table 3.

In Figure 8 and Table 3, in the hydroxyl end group side, the slope of the approximation was constant, whereas the slope of the approximation on the phenolic end group side was very large. This was responsible for the completely different behavior of *v_d_* in both end groups. At the phenyl end group, the reduction in *n_F_* was much greater than *n_C_*. This is due to the phenyl group of the terminal group, because the wavelength dependence of the molecular refractive index of the phenyl group is large [22,23].

However, the influence of the terminal becomes negligible as the molecular weight increases. Finally, the Abbe number of the 30-mers becomes 57 or more, regardless of the terminal.

We discuss why *CO* dominates *n_d_*. The refractive indices are described by the Lorentz-Lorenz equation below [24,25]:(7)$$n=\sqrt{\left(2\frac{\left[R\right]}{V}+1\right)/\left(1-\frac{\left[R\right]}{V}\right)}$$

In this equation, the refractive index is calculated by the molecular refractive index [*R*] and the molecular volume *V*. The refractive index of the phenolic end group DNDM-PC is enhanced by PhO groups in the small molecular range. However, the refractive index of the whole molecule does not increase largely because the effect of decreasing the molecular refractive index by the carbonyl groups is also large. As the molecular weight increases, the effect of the terminal group becomes smaller, and the effect by the carbonyl groups becomes dominant. On the other hand, the behavior of the refractive index at the hydroxy end group is also the same as the phenolic e_nd_ group. The larger the molecular weight, the more the carbonyl groups and the lower the refractive index. The behavior of decreasing refractive index becomes more pronounced as the molecular weight increases.

Therefore, it is considered that the refractive index of the phenolic end group behaves similarly to that of the hydroxy end group. These results indicate that *n_d_* is dominated by the carbonyl groups.

However, the *v_d_* does not show a similar trend. The phenolic end group is strongly affected by the short-wavelength F-line, resulting in an elevated refractive index. This effect is more pronounced for lower molecular weight bodies with higher end group concentrations. On the other hand, the formula for the *v_d_* below has the refractive index *n_F_* in the denominator.
*vd* = (*nd* − 1)/(*nF* − *nC*)(8)

Therefore, it is considered that the lower molecular weight body at the phenolic end group has a lower *v_d_*. The *n_F_* of the phenolic end group becomes less affected by the end when the molecular weight is sufficiently large and approaches the same value as the *v_d_* of the hydroxyl end group.

Since *v_d_* has a large molecular weight dependence and terminal end group, the setting of the region of molecular weight and the kind of the terminal end group is very important from the viewpoint of the optical stability of the products

### 3.3. Relation between T_g_ and D_p_

The relationship between *D_p_* and *T*_g_ is shown in Figure 9. As *D_p_* increases, the *T*_g_ increases. For example, in the case of the hydroxyl end series, *T*_g_ increases as 291.8 K (OH-1), 326.3 K (OH-2), 377.9 K (OH-8), 396.0 K (OH-14), and 411.2 K (OH-31). On the other hand, *T*_g_ increases as 299.2 K (Ph-0), 307.9 K (Ph-1), 368.7 K (Ph-8), 385.2 K (Ph-13), and 407.8 K (Ph-28-1) for the phenolic end series. At the same *D_p_*, the *T_g_* of the hydroxyl end series is higher than that of the phenolic end series. The difference in *T_g_* between the hydroxyl end series and the phenolic end series becomes smaller as *D_p_* increases. The reason for the higher *T_g_* of the hydroxyl end series is thought be due to the contribution of hydrogen bonding. The molecules in hydroxy end series have two hydroxyl groups in their molecular structure, which form intermolecular hydrogen bonds. As *D_p_* increases, the contribution of hydroxyl end group becomes smaller. Furthermore, the contribution of the hydroxyl end groups becomes negligible in the sufficiently large *D_p_* region.

Generally, the effect of molecular weight on *T_g_* is given by the well-known empirical equation, the Flory-Fox equation, Equation (9) [26,27,28]:*T_g_* = *T_g_*∞ − *K*/*M_n_*(9)

Applying the relationship between *M_n_* and *T_g_* for DNDM polycarbonate to this equation, it is found that the relationship can be well described in the polymeric domain, but the deviation in the oligomeric region is large (Figure 10).

Similarly, the relationships between *CO* that is derived from *D_p_* and 1/*T_g_* were examined, shown in Figure 11. Therefore, the conversion of *D_p_* to *CO* mentioned above is also applied to the analysis of *T_g_*. As a result, it was found that the relationship between the reciprocal of *T_g_* and *CO* can be fitted to the quadratic polynomial across the entire measured region, from the oligomeric to the polymeric region.

Since *CO* converges as *D_p_* increases, we can estimate the *T_g_* at infinite *D_p_* from the intersection of the two quadratic curves. Therefore, the *T_g_* of DNDM polycarbonate having infinite *D_p_* can be estimated as 421.9 K.

## 4. Conclusions

Previously, we only focused on polycarbonates with aromatic rings as monomer [16,17]; however, this time, we expanded the investigation to polycarbonates having alicyclic compound, DNDM, as monomer. It included the clarification of the influences of the *D_p_* and polymer end structures to the *n_d_*, *v*_d_, and *T_g_* for alicyclic polycarbonates.

For DNDM polycarbonates, the glass transition temperature increased with increasing *D_p_*, whereas the *n_d_* decreased as *D_p_* increased. In addition, both converged to a certain value when the *D*_p_ was sufficiently large. Interestingly, the tendency of *v*_d_ for change of *D_p_* was much different. Though the *v*_d_ is constant for the hydroxyl end series, in the case of the phenolic end series, the *v*_d_ was significantly affected by the change of *D_p_*.

This study revealed that optical properties such as *n_d_* and *v*_d_ were affected by the *D_p_* and polymer end structures. With the increasing demand for accuracy in optical properties, it is shown that we need to pay attention to perspectives that were previously thought to be unrelated to optical properties.

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
