# Peer review of "Correlation of the Abbe Number, the Refractive Index, and Glass Transition Temperature to the Degree of Polymerization of Norbornane in Polycarbonate Polymers"

_polymers, 2020, doi:10.3390/polym12112484_

Round 1

Reviewer 1 Report

The Article about Correlation of the Abbe number, the refractive index, and glass transition temperature to the degree of polymerization of Norbornane in Polycarbonate Polymers can be of interest to the reader of Polymer, but before being published the following major issues needs to be addressed

Figure 2 Generally the DSC is normalized by the weight of the sample and also it is indicated the direction for exo or endo transformation (depending on the positive or negative value of the y axis)

There is no indication in the paper about how many replicates were performed. Please add it and eventually were possible indicates the results reported including a standard deviations (See table 1 and table 2)

Figure 3 check the label of Figure 3

Figure 5, plase use sample indicators as in Figure 4 (weight percentage of 2,6,8 etc)

About the correlation between weight percent of the carbonyl groups CO and reflective index nd it is due to say that even if indeed a correlation exist it seems that there are two step  (6-9 and 9-11 % ranges)

This can probably be ascribed to the choice of the design of the experiment but without knowing the variance of the problem and due to the distribution of the points chosen it can be very misleading to rely on the coefficient obtained by a linear interpolation (that can vary a lot).  The author are suggested to report if possible the error bars on Figure 5 and eventually reperform a few replicates to check the validity of their interpolation. Also in order to better study the trend a sample with the lowest possible obtainable CO% should be measured (eventually confirming the results obtained  for the sample with 6%)

The same is true for the refractive index nC and Nf. Even if a second order interpolation seems more appropriate (but this should be checked with the consistency of the parameters considered for the mode) the authors can also think to better study the behavior of the system splitting the data in two intervals and creating two regression models

Even if it’s not wrong a priori, the authors use self citation for 1/4 of the references presented , Also expanding the reference section will improve the quality of the paper presented

Finally, I would remark that the topic is really interesting and it is refreshing reading a draft on a topic with such interesting practical outcomes.

Author Response

Dear Reviewer,

Thank you for your review to our manuscript.

It really helps us to see the phenomenon from the other viewpoint.

We'll response to your kindness with the attached file.

Sincerely

Authors

Reviewer 2 Report

   The manuscript is written in reasonable English, though some statements are incorrect or unclear, because of improper grammar or wording.  However, there is a problem with the experimental data work-up and interpretation.  The following is wrong in the manuscript:

  1. The main problem is that in all of Figures, the authors plot experimentally determined quantities (i.e., the refractive indices, Abbe number (νd) and the glass transition temperature  (Tg) or its reciprocal (1/Tg)) as a function of the degree of polymerization (Dp) or weight percentage of carbonyl groups (denoted as CO in italics), calculated theoretically on the basis of initial ratio of monomers before polymerization, instead of using real degree of polymerization (and the corresponding real percentage of carbonyls) that can be calculated from the real molecular weights (Mn) of the materials studied.  The Dp reported in Tables 1-2 represents only a hypothetical degree of polymerization that theoretically could be achieved at infinite reaction time if the polymerization proceeded cleanly without any side reactions,  while the real degree of polymerization values, that can be calculated from Mn , are much different.  For example, for the sample OH-100, with the theoretical degree of polymerization equal 100, the average  Dp determined from Mn = 7587 g/mol is 30.6, while that of the sample Ph-100 is 28.1.  Real experimental data for Dp and CO should be used for the plots.
  2. lines 189-192:  More exactly: the refractive index of low molecular weight polymers (usually called "oligomers") depends slightly on Dp, because nd depends on the material structure, while only at low molecular weights the contribution of partial molar refractivity of terminal groups to the overall refractivity of the polymer is significant. 
  3. The denominators in the expressions shown in lines 196 and 205 represent molecular weight of the corresponding polymers.  So, those expressions can be used for determination of the real Dp values of the samples studied from the real molecular weights (Mn) reported in Table 2.
  4. Figure 4 is simply a plot of the mathematical functions represented by the equations shown in the lines 196 and 205.  So, it may be plotted using lines instead of points  (i.e., the data for any intermediate Dp values also can be calculated, so there is no excuse to plot only 3 data points for the Dp range above 30).
  5. For the OH-terminated polymer series, the dependencies of the refractive indices on the percentage of carbonyl groups, shown in Figures 5 and 7, are only approximately linear – second order polynomial would fit much better.  Hence,  the statement in line 228 should contain "...had an approximately linear correlation as shown in Figure 5 and was expressed..." to be more precise.
  6. lines 241-247:  This discussion makes no sense, because the authors first call the O(C=O)O group a "linker" and next they  specify that "...the molecular weight of the linker part becomes close to a specific limit (critical molecular weight)."  The molecular weight of the O(C=O)O linker equals 60 and is constant. 
  7. The statements in the lines 248-253 also make no sense, because  Dp of polymers obtained by condensation polymerization could only theoretically go to infinity at exactly 1:1 ratio of the monomers, after infinitely long reaction time and in the absence of any side reactions that change the monomer ratio during the polymerization process.  Moreover, the properties of high molecular weight polymers (i.e., those with Dp > 100) can be measured directly without need for the intersection method proposed by the authors for oligomers.
  8. line 254: The change of refractive index by only 0.02 (as shown in Figure 3) upon change of molecular weight of an oligomer from 287 g/mol (sample Ph-2) to 7195 g/mol (sample Ph-100) is hard to call "a large molecular weight dependence".
  9. The Abbe numbers reported in Table 1 and plotted in Figure 6 are not consistent with the Abbe numbers calculated from the refractive indices reported in that Table using the equation shown in line 310.  The nd data in Table 1 seem to be smoothed out to make the data for OH-terminated polymers shown in Figure 6 less scattered.  Smoothing experimental data and reporting them as real data is not acceptable for any scientific journal.
  10. lines 271-272: The statement: "And refractive indices were calculated based on CO in Table 3." is incorrect, because the refractive indices were measured experimentally, not calculated.
  11. lines 293-308: This part of the discussion is unclear and speculative, because the authors use the same symbol (i.e., italic "CO") for denoting weight percentage of carbonyl groups in the polymers and for the carbonyl group itself.
  12. line 343: "We discuss why CO dominates Tg" – It is not clear what is meant by that (i.e., what "CO"? carbonyl groups or their contents in the polymers?)
  13. lines 349-352: The statement: "Since the molecular weight Mn and CO are the same dimension, and the same units; g/mol, it is directly considered that 1/Tg is dependent on CO." does not make sense, because the Mn is expressed in [g/mol], while CO was expressed in [%].
  14. lines 351-352: Also the statement: "We can describe the equation above to be the same as the one below." makes no sense, because the reciprocal of a sum of two components is not equal to the sum of reciprocals of the components. Moreover, the relationship in line 354 is a purely empirical description of the data plotted in Figure 9 without any physical sense, because the 1/Tg was expressed in [oC-1] instead of using absolute temperature units [K].
  15. Finally, the word "greatly" in the Conclusion section is not used properly, because only Tg depends greatly on Dp, while the relative changes of refractive indices are small (i.e., less than 2% of the refractive index values).

Moreover, the following improper wording needs to be corrected:

- Figure 3 titlethere is: reflective --> there should be: refractive

- line 212: phenylen ring --> phenoxy group (or: benzene ring, because this is not "phenylen")

- line 295: PhOH --> PhO groups

Author Response

(The authors gave the same response as above.)

Reviewer 3 Report

The work by N. Kato et al. study the influence of polymerization degree upon the Abbe number, refractive index and Tg. The work is very important for the precise application of  Norbornane in Polycarbonate Polymers. But there remain some issues. I can’t recommend it till all my concerns below have well been responded:

  1. What is the logic link between the first two paragraphs? It is too abrupt.
  2. The motivation is not clear. Why do you study with Dp? Other parameters can be more important. Any reason or design consideration for it?
  3. Please give out the specific temperature and pressure used for hydrogenation.
  4. Please define the symbols, like nd, nc and nF before the citation. What is the difference among them?
  5. For the sample for index test, should they need be polished? If so, please give more information about them.
  6. For all the linear fittings, please add the error bar.
  7. Why the results in Table 3 for Phenolic end group get some much difference? How does it explan by the material dispersion together with the dispersion spectrum of phenolic end group.
  8. The equation for the relationship between Abbe number and refractive index should be give out the citation, as well as Flory-Fox equation. In addition, it is better to number all the equations in the paper.
  9. The English is causal at some places. It is better to have thorough and careful check and improvement.

Author Response

(The authors gave the same response as above.)

Round 2

Reviewer 2 Report

   The authors have corrected most of the meritorius errors I pointed out in my previous review satisfactorily enough, but they miscalculated the real degree of polymerization values (Dp  in Tables 1 and 2) on the basis of real molecular weights (Mn in Table 2) of the oligomers studied.  Namely, the authors calculated Dp by division of Mn by the molecular weight of DNDM monomer, which is totally wrong, because such a ratio can be used only for approximate calculation of the degree of polymerization of high molecular weight polymers obtained by addition polymerizations, where no low-molecular weight side product is released during the polymerization process and where contribution of terminal groups to the molecular weight can be neglected.   It cannot be used for oligomers nor for polymers obtained by condensation polymerization.  The oligomers studied by the authors were obtained by condensation polymerization of DNDM monomer with diphenyl carbonate (DPC in Scheme 2) with release of phenol (PhOH) as the side product. 

    In the Comment 3 in my previous review, I provided a hint on how correct and accurate degree of polymerization (Dp) values can be calculated from the number-average molecular weights (Mn), but that remark was ignored.  Consequently, all the Dp data in Tables 1 and 2 have to be recalculated in the following way:

   As I specified in Comment 3 from my previous review, the denominators in the equations used for calculation of the weight percentage of carbonyl groups (CO) in the polymers studied (now labeled as equations 2 and 3) represented the molecular weight of the polymers. Hence:

For the hydroxy-terminated polymer samples:

   Mn = bDp + a(Dp-1)  - 2c(Dp-1)

 For the phenoxy-terminated polymer samples:

   Mn = bDp + a(Dp+1)  -  2cDp + 2d

   So, it is enough to rearrange those equations to get the following accurate expressions for the calculation of average degrees of polymerization (Dp) on the basis of experimental Mn values of the polymer samples studied:

For the OH-terminated samples:

  Dp = (Mn + a – 2c)/(b + a – 2c)

For the PhO-terminated samples:

  Dp = (Mn - a – 2d)/(b + a – 2c)

where:

a = 28.01 g/mol (molecular weight of carbonyl group),

b = 222.33 g/mol (molecular weight of DNDM monomer),

c = 1.01 g/mol (molar mass of hydrogen atom),

d = 93.11 g/mol (molecular weight of phenoxy group).

Next, correct values of the real percentage of carbonyl groups (CO) in the samples studied need to be recalculated on the basis of the corrected Dp values, using the equations (2) and (3), and all of the figures where Dp or CO data were used (i.e., Figures 3,5,6,7,8,9 and 11) need to be redrawn based on the corrected data.

Author Response

Dear Reviewer,

Thank you for your advice, which made the manuscript so clear.

We have fixed them as your advice and really appreciated for them.

Thank you very much again.

Best Regards

Authors,

Reviewer 3 Report

All my concerns have well been responded and I recommend the acceptance for the publication. 

Author Response

Dear Reviewer,

Thank you for your reviewing our manuscript.

Your advice was so helpful for revising it. 

Best regards

Authors,